# The effect of job and personal demands and resources on healthcare workers' wellbeing: A cross-sectional study

Consuela Cheriece Yousef[1,2,3☯]*, Ali Farooq[4,5☯], Gigi Amateau[6‡], Laila Carolina Abu Esba[7,8,9‡], Keisha Burnett[10‡], Omar Anwar Alyas[11‡]

1 Pharmaceutical Care Department, Ministry of National Guard-Health Affairs, Dammam, Saudi Arabia, 2 King Abdullah International Medical Research Center, Al Ahsa, Saudi Arabia, 3 King Saud bin Abdul-Aziz University for Health Sciences, Al Ahsa, Saudi Arabia, 4 Department of Computer and Information Sciences, University of Strathclyde, Glasgow, United Kingdom, 5 Qatar Computing Research Institute, Hammad bin Khalifa University, Doha, Qatar, 6 Department of Gerontology, College of Health Professions, Virginia Commonwealth University, Richmond, Virginia, United States of America, 7 Pharmaceutical Care Department, Ministry of National Guard-Health Affairs, Riyadh, Saudi Arabia, 8 King Abdullah International Medical Research Center, Riyadh, Saudi Arabia, 9 King Saud bin Abdul-Aziz University for Health Sciences, Riyadh, Saudi Arabia, 10 Department of Clinical Laboratory Sciences, Cytopathology Practice Program, University of Tennessee Health Science Center, Memphis, Tennessee, United States of America, 11 College of Medicine, Royal College of Surgeons in Ireland—Medical University of Bahrain, Kingdom of Bahrain

☯ These authors contributed equally to this work.
‡ GA, LCAE, KB and OAA also contributed equally to this work.
* yousefco@mngha.med.sa

**Data Availability Statement:** All relevant data are within the paper.

**Funding:** The authors received no specific funding for this work.

## Abstract

The COVID-19 pandemic presented many psychological stressors which affected healthcare worker wellbeing. The aim of this study was to understand the factors that affect the wellbeing of healthcare professionals in the Kingdom of Saudi Arabia using Job-Demand and Resource (JD-R) Model. The proposal model consisted of demand factors (Work load—job demand, loneliness—personal demand), support factors (organizational support—job resource, and resilience—personal resource), mediators (burnout and work engagement), and outcome (wellbeing) A cross-sectional, descriptive study was conducted across 276 healthcare workers from hospitals and primary healthcare centers, including healthcare professionals, health associate professionals, personal care workers, health management and support personnel, and health service providers, and others between February-March 2022. The proposed model was tested using partial least squares structural equation modeling. Among the respondents, the majority were female (198, 71,7%), married (180, 65.2%), healthcare professionals (206, 74.6%), being more than 10 years in the profession (149, 51.6%), and non-Saudi nationality (171, 62.0%). Burnout accounted for a significant effect on wellbeing. Of the demands (workload and loneliness) and the resources (organizational support and resilience), workload had the greatest impact on burnout. Healthcare organizations should invest in reducing workloads and promoting resilience to reduce burnout and increase healthcare worker wellbeing.

**Competing interests:** The authors have declared that no competing interests exist.

## Introduction

In January 2020, the World Health Organization declared COVID-19 a public health emergency of international concern [1]. With the COVID-19 global pandemic, there were unprecedented changes in public health policies and healthcare systems to respond to the challenges. Travel restrictions, lockdowns, quarantines, strict social distancing, rationing of healthcare supplies, and the implementation of telehealth were a few of the changes that occurred at a rapid pace in effort to control the spread of the virus and care for the ill [2, 3]. The consequences were widespread, affecting the cultural, social, political, and economic landscapes around the world [4].

As healthcare systems struggled to maintain control and implement preventive strategies while conserving resources, healthcare workers were particularly affected by the changes. A number of studies evaluated the psychological effects experienced by healthcare workers [4–10]. Studies found high rates of anxiety, burnout, insomnia, stress, post-traumatic stress disorder, and depression during the pandemic [8, 10, 11]. This was due to risks associated with contracting the virus, fear of death from the disease, stigmatization, and social isolation due to safety precautions [8, 12, 13]. Financial hardships and stress related to uncertainty regarding the continued impact of the pandemic also contributed to the psychological burden [8]. Additionally, healthcare workers worked long hours and endured heavy workloads [14].

With the implementation of social distancing policies, concerns were raised about increased loneliness due to a lack of engagement with teleworking and impaired connections with families and loved ones causing social isolation [15, 16]. Researcher Julianne Holt-Lunstad has extensively studied the effects of loneliness and social isolation on emotional and physical wellbeing [15–20]. A lack of social connection has been described as "a risk that is comparable, and in many cases, exceeds that of other well-accepted risk factors, including smoking up to 15 cigarettes per day, obesity, physical inactivity, and air pollution"[17].

Prior to the pandemic, loneliness was identified as a public health concern in the United Kingdom and the United States due to its association with increased morbidity and mortality [21, 22]. Subsequently, in 2023, the U.S. Surgeon General released a health advisory raising impaired social connection as a public health crisis in the United States [23].

Researchers found a significant prevalence of loneliness (21%, pandemic vs 6%, pre-pandemic) among general populations around the globe during the pandemic [24–27]. Healthcare workers, in particular, were impacted by the pandemic, with studies showing a higher prevalence than in the general population in some locations (53%, Spanish healthcare workers [28]; 89%, Bangladeshi healthcare workers [29]; 60%, Australian healthcare workers [30]). No studies have been found pertaining to loneliness during the pandemic among healthcare workers in Saudi Arabia, specifically.

Research has been conducted on the impact of loneliness and other psychological stresses on healthcare workers and their wellbeing during the pandemic [14, 24, 31]. Job characteristics, including work load and organizational support, also impact employee wellbeing. In 2016, a systematic review evaluated the association between healthcare staffs' wellbeing and burnout with patient safety and found poor wellbeing was correlated with more patient safety issues [32]. Therefore, understanding the factors that foster employee wellbeing in the healthcare setting will help policymakers to provide a supportive environment for high-quality, safe care.

## Aim of the study

The aim of this study was to analyze the relationship of job demands (i.e., work load), personal demands (i.e., loneliness), job resources (i.e., organizational support), and personal resources (i.e., resilience) on burnout, work engagement, and wellbeing in a population of healthcare

workers in an integrated healthcare system in the Kingdom of Saudi Arabia. There was a high prevalence of psychological distress reported by healthcare workers during the pandemic [33]. To promote efficient, high-quality healthcare, a better understanding of how these factors work together is needed.

## Theoretical framework and hypotheses

The Job Demands-Resources (JD-R) theoretical framework was initially developed to understand how stress from job demands and motivation from job resources in occupational environments can predict burnout and work engagement [34–36]. It has been used to examine predictors of worker outcomes such as wellbeing, engagement, and individual- and organizational-level outcomes [37]. Schaufeli and Taris recommended using the JD-R model to improve employee wellbeing by tailoring it to the specific situation and needs of the organization, incorporating negative and positive outcomes and processes for a more balanced approach [38]. Many researchers have used JD-R as a framework to evaluate various relationships among healthcare workers [9, 37, 39–47]. Job demands and job resources are "physical, psychological, social, or organizational aspects of the job" [34]. Job demands can create stress and strain on individuals over time. Job resources can reduce job demands and lead to work engagement [34]. Personal demands and personal resources were later additions to the JD-R model [27, 48, 49]. Personal demands are "individual characteristics that are reflected in employees' effort in their work" [27]. Loneliness has been conceptualized as a personal demand during COVID-19 [27]. Personal resources are positive individual characteristics associated with resiliency and the ability to navigate the environment successfully [49]. Subjective wellbeing has been used broadly to capture several phenomena, including emotional responses, satisfaction with health and work, and overall satisfaction with life [50].

Studies have shown resilience, an individual characteristic of coping and adaptability, to enhance psychological wellbeing [2, 51, 52]. Resilience has been identified as a protective factor to counter an individual's exposure to challenging situations [5]. In healthcare workers, resilience was found to be a significant predictor of psychological wellbeing [53]. There is interindividual variation in resilience, with higher levels of resilience associated with fewer physical ailments (i.e, headaches, musculoskeletal pains), lower levels of depression, better relationships, and less concern for environmental stimuli [5]. Resilience factors mitigate the consequences of the mental distress that occur following a negative experience and include such characteristics as: confidence, positive mental state, ability to adapt and cope, a strong sense of purpose, optimism, and perception of strong social support [52]. Some resilience factors are fixed characteristics, whereas others are modifiable. Workplace cultures and systems can improve individual resilience by buffering work stressors [2].

This study used the JD-R model to examine how job demands (workload), personal demands (loneliness), job resources (organizational support), and personal demands (resilience) relate to burnout, work engagement, and employee wellbeing (Fig 1).

## Materials and methods

### Study setting and design

A cross-sectional, descriptive study was conducted at the primary care facilities and hospitals of the Ministry of National Guard Health Affairs (MNGHA) in the cities of Dammam, Al Ahsa, Riyadh, Madinah, and Jeddah.

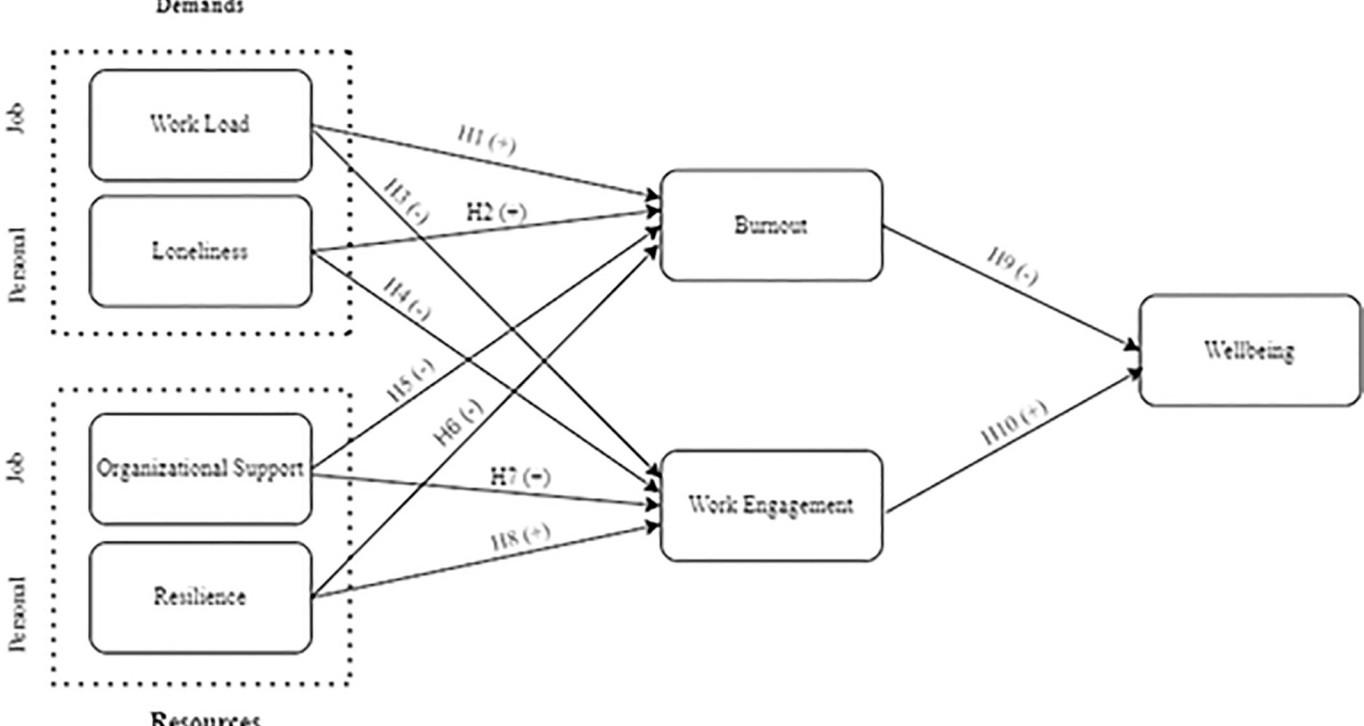

**Fig 1. Adapted JD-R model with hypotheses.**
- H1: There will be a positive relationship between workload and burnout.
- H2: There will be a positive relationship between loneliness and burnout.
- H3: There will be a negative relationship between burnout and wellbeing.
- H4: There will be a negative relationship between workload and work engagement.
- H5: There will be a negative relationship between loneliness and work engagement.
- H6: There will be a positive relationship between organizational support and work engagement.
- H7: There will be a positive relationship between resilience and work engagement.
- H8: There will be a negative relationship between organizational support and burnout.
- H9: There will be a negative relationship between resilience and burnout.
- H10: There will be a positive relationship between work engagement and wellbeing.

## Study participants

Healthcare workers were selected from hospitals and primary healthcare centers under any MNGHA facility. The categories for healthcare workers were taken from the World Health Organization and included healthcare professionals (e.g., physician, dentist, pharmacist, nurse, occupational/physical/respiratory therapist), health associate professionals (e.g., medical imaging technician, pharmacy technician, dental technician, optician, emergency medical technician), personal care workers (e.g., patient care aide, dental aide, pharmacy aide, phlebotomist), health management and support personnel (e.g., managerial staff, social workers clerical workers, human resources, information and technology), and health service providers not elsewhere classified (e.g., members of the armed forces, student intern, and hospital volunteer) [54]. To ensure an adequate sample size for performing partial least squares structural equation modeling for the statistical analysis, the target sample size was 200 [55]. Convenience sampling was used to recruit participants from all cities.

## Ethical approval

This study received IRB approval from the King Abdullah International Medical Research Center. The first page of the survey contained information about the study and enabled participants to provide informed consent. Participation in the study was voluntary, responses were anonymous, and no personal information was collected.

## Recruitment

Data were collected through an anonymous online survey in English using Google Forms between February and March 2022. The survey link was shared through email invitations sent through the organization's listserv for healthcare workers in all cities (Dammam, Al Ahsa, Riyadh, Madinah, and Jeddah) and through WhatsApp messages sent to personal contacts of two of the authors. Reminder emails and WhatsApp messages were sent at one-week intervals. We could not calculate the response rate because it is difficult to determine the number of healthcare workers who received the invitation.

## Instrument

The instrument was pilot tested with seven healthcare workers working within MNGHA. A Google Forms survey link and a cover letter explaining the purpose of the study were emailed to the HCPs to obtain feedback regarding the questionnaire's length, clarity, and flow. After comments were compiled, five items were removed, and some were modified to improve clarity and decrease survey length. The final version of the survey included 46 items which covered nine sections: demographic characteristics; job demands; personal demands; job resources; personal resources; burnout; work engagement; subjective wellbeing; and three open-ended questions (to be analyzed separately) soliciting additional comments and concerns about their experiences working through the pandemic.

Eleven items were collected: profession, main worksite, city, work with patients who had COVID-19, years of experience, nationality, marital status, gender, age, housing situation, and time away from family Table 1.

## Measures

The main measures in the instrument were taken from the earlier studies and adapted where required to match the study's needs and are described below. All were measured on a five-point scale (1 = strongly disagree, 5 = strongly disagree).

**Job demands.** Job demands (JD) included workload. The workload was measured by three items [56], which were:

JD1. During the COVID pandemic, there are more demands at work.

JD2. Since the pandemic, I feel like I have a lot to do at work.

JD3. My job requires so much from me now.

**Personal demands.** Personal demands included loneliness. Loneliness was measured by three items [27], which were:

PD1. Since the COVID pandemic, I often feel lonely.

PD2. I frequently feel left out since the COVID pandemic.

PD3. During the pandemic, I regularly feel isolated from others.

**Table 1. Demographic characteristics (N = 276).**

| Variables | Respondents, n (%) |
|---|---|
| Healthcare facility | |
| Dammam | 47 (17) |
| Madinah | 31 (11.2) |
| Al Ahsa | 69 (25.0) |
| Jeddah | 107 (38.8) |
| Riyadh | 22 (8.0) |
| Type of facility | |
| Hospital | 261 (94.6) |
| Primary healthcare clinic | 12 (4.3) |
| Gender | |
| Female | 198 (71.7) |
| Age | |
| 20–29 years | 32 (11.6) |
| 30–39 years | 112 (40.6) |
| 40–49 years | 86 (31.2) |
| More than 50 years | 43 (15.6) |
| Marital status | |
| Married | 180 (65.2) |
| Housing situation | |
| Live alone | 58 (21.0) |
| Live with your family | 136 (49.3) |
| Shared accommodation with non-relative | 80 (29.0) |
| Healthcare worker | |
| Health professional | 206 (74.6) |
| Health associate professional | 17 (6.2) |
| Health management and support personnel | 32 (11.6) |
| Personal care worker | 11 (4.0) |
| Other | 7 (2.5) |
| Years in profession | |
| Less than 5 years | 31 (11.2) |
| 5–10 years | 64 (23.2) |
| More than 10 years | 180 (65.2) |
| Years working at MNGHA | |
| Less than 1 year | 23 (8.0) |
| 1–4 years | 43 (14.9) |
| 5–10 years | 74 (25.6) |
| More than 10 years | 149 (51.6) |
| Nationality | |
| Saudi | 103 (37.3) |
| Non-Saudi | 171 (62.0) |
| Time away from family | |
| Less than 6 months | 46 (16.7) |
| 6 months-1 year | 46 (16.7) |
| More than 1 year | 64 (23.2) |
| Not applicable | 120 (43.5) |

Note: N donates sample size, whereas, n donates sub-samples in each group

**Job resources.**   Job resources included organizational support. Organizational support (OS) was measured by four items [57], which were:

OS1. I have access to appropriate personal protective equipment (masks, gloves, gowns).

OS2. I am certain my organization would take care of my needs if I become infected with the COVID-19 virus.

OS3. As work demands increase, I can get the support I need from the organization.

OS4. During the COVID-19 pandemic, I have had access to up-to-date information and communication from the organization.

**Personal resources.**   Personal resources included resilience. Resilience (R) was measured using the Brief Resilience Scale [58] using six items, as below:

R1. I tend to bounce back quickly after hard times.

R2. I have a hard time making it through stressful events.

R3. It does not take me long to recover from a stressful event.

R4. It is hard for me to feel like my normal self.

R5. I usually come through difficult times with little trouble.

R6. I tend to take a long time to get over set-backs in my life.

**Burnout.**   Burnout (BO) was measured with the Copenhagen Burnout Inventory [59]. The seven items were:

BO1. Since the pandemic, work is emotionally exhausting.

BO2. I feel burned out due to my work during the COVID-19 pandemic.

BO3. My work frustrates me quite a bit since the pandemic.

BO4. I frequently feel worn out at the end of the working day since the COVID-19 pandemic.

BO5. During the pandemic, I feel exhausted in the morning at the thought of another day at work.

BO6. Every working hour is tiring for me since the COVID-19 pandemic.

BO7. I never have enough energy for family and friends during my leisure time since the COVID-19 pandemic.

**Work engagement.**   Work engagement (WE) was measured with the Utrecht Work Engagement Scale-3 instrument [60]. The three items were:

WE1. At my work, I feel bursting with energy since the COVID-19 pandemic.

WE2. I am enthusiastic about my job during the pandemic.

WE3. I am immersed in my work since the pandemic.

**Subjective wellbeing.**   Subjective wellbeing (WB) was measured with six items [61] stated below:

WB1. I feel that life is meaningful.

WB2. I enjoy my life.

WB3. I hold goals and beliefs that affirm a sense of direction in life, and I feel that life has purpose and meaning.

WB4. In the last three months, I have had difficulty sleeping.

WB5. My mental health is generally excellent.

WB6. My physical health is generally excellent.

### Data analysis

Descriptive statistics were obtained using SPSS version 25 [62]. Partial least squares structural equation modeling (PLS) was used to test the research model using SmartPLS Version 3.0 [63]. The reliability of the measurement model was evaluated with the composite reliability (CR). A CR greater than 0.708 was indicative of construct reliability. The factor loadings and average variance extracted (AVE) were examined to determine convergent validity. Indicator loadings were required to be greater than 0.7 and AVE values greater than 0.5 [64]. The Fornell-Larcker criterion, wherein the square root of the AVE of each construct should be greater than its highest correlation with any other construct, was used to evaluate discriminant validity [64]. Indicators were removed if the variance inflation factor was more than five, indicating collinearity or if the weight and the loading were insignificant. This analysis approach has been used extensively in healthcare [11, 65] and other disciplines, such as information systems [66], tourism [67], education [68], and cybersecurity [69].

### Results

### Demographics

The characteristics of the 276 respondents Table 1. Most participants were health professionals (74.6%), female (71.7%), 30–39 years of age (40.6%), and non-Saudi (62.0%). The majority had greater than 10 years' experience (65.2%) and were hospital-based (94.6%). In terms of housing situation, 21% were living alone while 49% were living with family, and 29% were in shared accommodation.

### Measurement model test statistics

Table 2 shows the measurement model test statistics, which provide evidence of the quality of the constructs involved. Of 32 items, seven items—OS1, OS4, R1, R3, R5, WE3, and WB3—had an item loading less than 0.6 and were removed from the analysis. The composite reliability of all constructs was between 0.79 and 0.93, and AVEs were higher than 0.50. One construct, work engagement, had a Cronbach alpha value less than the threshold of 0.6; however, given composite reliability and AVE appropriate, this anomaly did not affect the quality of the constructs. Moreover, the variance inflation factor (VIF) of all items was less than 5, showing a lack of multicollinearity among the items. In addition, the construct level VIF was less than 3.3, providing evidence for the lack of common method bias in the dataset.

Table 3 shows the discriminant validity of the constructs. Discriminant validity was evaluated utilizing the Fornell-Larcker criterion. The square roots of the AVE are presented in italics, demonstrating that each construct's AVE surpassed its maximum correlation with any other construct.

**Table 2. Measurement model statistics.**

| Construct/Items | Mean | SD | VIF | Loadings | Alpha | CR | AVE |
|---|---|---|---|---|---|---|---|
| Workload | 4.11 | 0.96 | 1.247 | | 0.84 | 0.90 | 0.77 |
| WL1 | 4.27 | 1.06 | 2.007 | 0.83 | | | |
| WL2 | 3.99 | 1.13 | 1.97 | 0.89 | | | |
| WL3 | 4.07 | 1.09 | 2.354 | 0.90 | | | |
| Loneliness | 2.98 | 1.15 | 1.960 | | 0.83 | 0.90 | 0.75 |
| L1 | 2.92 | 1.33 | 2.145 | 0.89 | | | |
| L2 | 2.79 | 1.33 | 2.002 | 0.89 | | | |
| L3 | 3.22 | 1.31 | 1.791 | 0.82 | | | |
| Organizational Support | 3.18 | 1.09 | 1.066 | | 0.70 | 0.87 | 0.77 |
| *OS1* | 4.22 | 0.99 | *1.195* | *0.51* | | | |
| OS2 | 3.47 | 1.26 | 1.407 | 0.84 | | | |
| OS3 | 2.89 | 1.22 | 1.407 | 0.90 | | | |
| *OS4* | 3.82 | 1.13 | *1.172* | *0.52* | | | |
| Resilience | 2.93 | 1.04 | 2.023 | | 0.78 | 0.87 | 0.69 |
| *R1* | 3.28 | 1.13 | *1.252* | *0.14* | | | |
| R2$^r$ | 2.88 | 1.26 | 1.588 | 0.86 | | | |
| *R3* | 3.22 | 1.17 | *1.195* | *0.38* | | | |
| R4r | 2.88 | 1.31 | 1.641 | 0.81 | | | |
| *R5* | 3.12 | 1.08 | *1.193* | *-0.39* | | | |
| R6r | 3.05 | 1.18 | 1.581 | 0.80 | | | |
| Burnout | 3.54 | 1.04 | 1.129 | | 0.91 | 0.93 | 0.67 |
| BO1 | 4.05 | 1.16 | 1.900 | 0.77 | | | |
| BO2 | 3.64 | 1.28 | 3.118 | 0.87 | | | |
| BO3 | 3.33 | 1.32 | 2.324 | 0.80 | | | |
| BO4 | 3.58 | 1.22 | 2.345 | 0.82 | | | |
| BO5 | 3.42 | 1.34 | 2.85 | 0.86 | | | |
| BO6 | 3.36 | 1.27 | 2.636 | 0.83 | | | |
| BO7 | 3.40 | 1.31 | 1.845 | 0.75 | | | |
| Work Engagement | 3.27 | 0.97 | 1.052 | | 0.54 | 0.79 | 0.65 |
| WE1 | 3.17 | 1.16 | 1.164 | 0.641 | | | |
| WE2 | 3.36 | 1.17 | 1.164 | 0.952 | | | |
| *WE3* | 2.50 | 1.01 | *1.034* | *-0.182* | | | |
| Wellbeing | 3.59 | 0.91 | | | 0.80 | 0.86 | 0.56 |
| WB1 | 4.00 | 1.13 | 1.863 | 0.669 | | | |
| WB2 | 3.61 | 1.20 | 1.795 | 0.788 | | | |
| *WB3* | 4.04 | 0.98 | *1.485* | *0.511* | | | |
| WB4r | 3.04 | 1.45 | 1.263 | 0.682 | | | |
| WB5 | 3.70 | 1.21 | 1.972 | 0.802 | | | |
| WB6 | 3.57 | 1.14 | 1.923 | 0.798 | | | |

Note: SD: Standard deviation, VIF: Variance inflation factor, CR: Composite reliability, AVE: Average variance extracted.

## Hypotheses testing

Structural model test results were used for hypotheses testing. Workload, loneliness, organization support, and resilience were antecedents, while burnout and work engagement were mediators. Wellbeing was the dependent variable. The factors gender, age, nationality, marital

**Table 3. Correlation matrix and square roots of AVE shown in the diagonal (p <0.05).**

| | 1 | 2 | 3 | 4 | 5 | 6 | 7 |
|---|---|---|---|---|---|---|---|
| 1. Burnout | *0.818* | | | | | | |
| 2. Workload | 0.646 | *0.876* | | | | | |
| 3. Organizational Support | -0.373 | -0.159 | *0.876* | | | | |
| 4. Loneliness | 0.638 | 0.397 | -0.185 | *0.867* | | | |
| 5. Resilience | -0.649 | -0.413 | 0.239 | -0.689 | *0.83* | | |
| 6. Wellbeing | -0.596 | -0.253 | 0.427 | -0.518 | 0.576 | *0.715* | |
| 7. Work Engagement | -0.098 | 0.056 | 0.32 | -0.124 | 0.063 | 0.371 | *0.812* |

status, housing situation, and exposure to COVID-19 patients were control variables for the dependent variable. The hypotheses testing results are shown in (Fig 2) and Table 4.

From the data, we found a positive relationship between workload and burnout (β = 0.40, t = 9.084, p <0.01). This means an increased workload is associated with increased burnout, with an effect size of 0.398. Loneliness was also found to positively impact burnout (β = 0.27, t = 5.547, p <0.01) with an effect size of 0.113 (H2). Workload has a large effect size in comparison to loneliness, having medium effect size [70].

The negative relationship between workload and work engagement, as hypothesized in H3, could not be established (β = 0.14, t = 1.960, p = 0.05). However, the hypothesis of a negative relationship between loneliness and work engagement (H4) (β = -0.17, t = 2.096, p <0.05) with an effect size of 0.017 was supported.

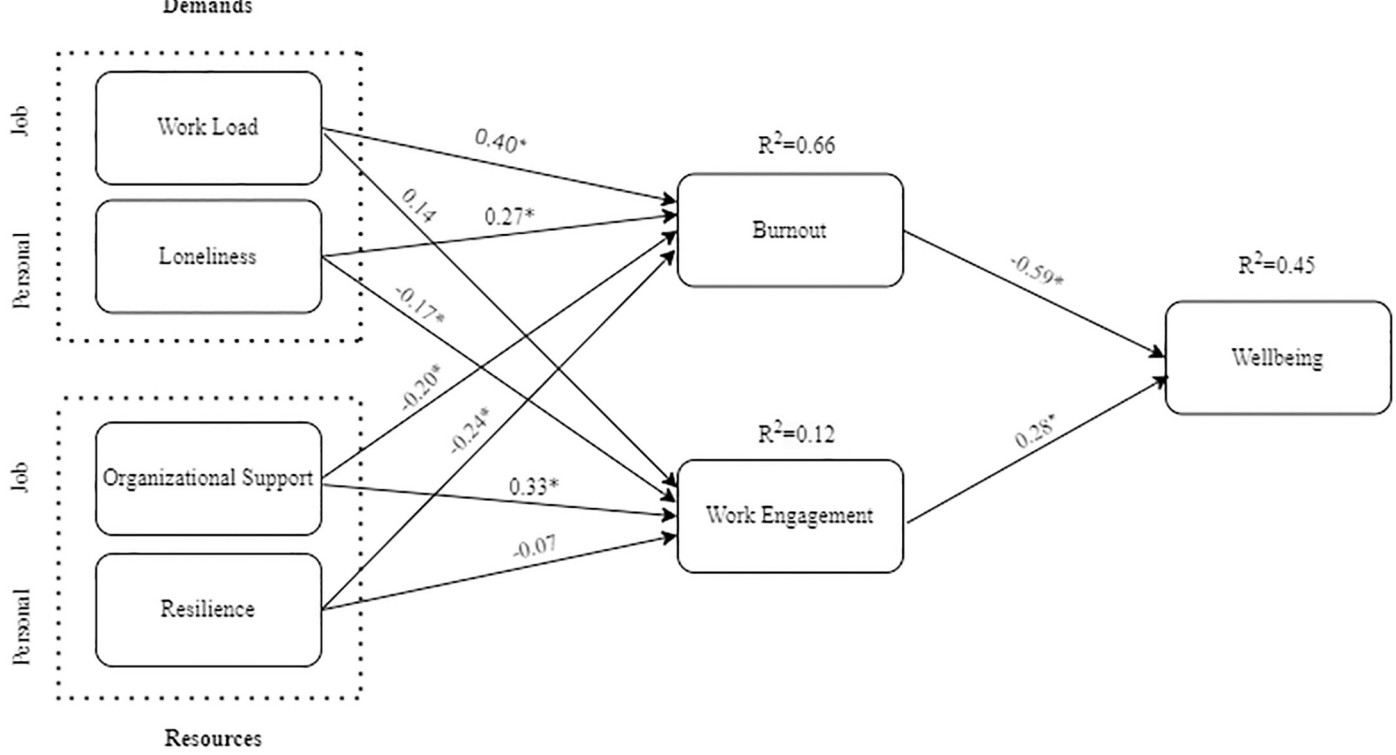

**Fig 2. Structural model test results.**

**Table 4. Hypotheses testing results.**

| Hypotheses | Relationship | Beta | t statistics | p | f2 |
|---|---|---|---|---|---|
| H1 | WL → BO | 0.40 | 9.084 | <0.01 | 0.398 |
| H2 | LON → BO | 0.27 | 5.547 | <0.01 | 0.113 |
| *H3* | *WL → WE* | *0.14* | *1.960* | *0.051* | *0.019* |
| H4 | LON → WE | -0.17 | 2.096 | 0.037 | 0.017 |
| H5 | OS → BO | -0.20 | 5.919 | <0.01 | 0.112 |
| H6 | RES → BO | -0.24 | 5.030 | <0.01 | 0.092 |
| H7 | OS → WE | 0.33 | 5.339 | <0.01 | 0.118 |
| *H8* | *RES → WE* | *-0.07* | *0.784* | *0.433* | *0.003* |
| H9 | BO → WB | -0.59 | 14.373 | <0.01 | 0.594 |
| H10 | WE → WB | 0.28 | 4.890 | <0.01 | 0.141 |

Note: WL = Workload, BO = Burnout, WB = Wellbeing, WE = Work Engagement, LON = Loneliness, OS = Organizational Support, RES = Resilience. $f^2$ shows the effect size. Unsupported hypotheses are shown in italic

We proposed two hypotheses to understand the resources' impact on burnout. The negative relationship between organizational support and burnout (H5) was supported: (β = -0.20, t = 5.919, p <0.05). This means that an increase in organizational support is related to a decrease in burnout, with an effect size of 0.112. H6 proposing a negative relationship between resilience and burnout was also supported: (β = -0.24, t = 5.0304 p <0.05). The effect size was 0.092.

To understand the impact of resources on work engagement, two hypotheses (H7 and H8) were proposed. The structural model results showed a significant positive relationship between OS and WE (β = 0.33, t = 5.339, p <0.01) with an effect size of 0.118. In the case of H8, no significant relationship was found between resilience and work engagement (β = -0.07, t = 0.784, p = 0.43).

Finally, the negative relationship between burnout and wellbeing (H9) was also supported: (β = -0.59, t = 14.373, p <0.01). This implies that an increase in burnout is associated with a decrease in work engagement, with an effect size of 0.594. We also found support for the positive relationship between work engagement and wellbeing (β = 0.28, t = 4.890, p <0.01). The effect size for this relationship was 0.141.

## Discussion

### Key findings

First, we found that workload and loneliness, demand factors from job and personal domain respectively, both affected negatively on the healthcare professionals' wellbeing through creating feeling of burnout. Both demand factors gave rise to burnout that negatively affected wellbeing. However, workload played a more significant role in burnout than did personal loneliness. In line with findings from studies conducted among physicians, higher workloads were positively associated with burnout [45, 71]. Studies among Belgian nurses [47] and Norwegian healthcare workers [37] also found workload to be positively associated with burnout.

Second, among the demand factors, only loneliness, the personal demand factor, significantly affected work engagement among the healthcare professionals. Studies showed mixed results on the association between loneliness and work engagement [72]. With respect to workload, the work-related demand factor, there was no significant effect on work engagement. However, in an integrative literature review, the authors found workload to be a predictive factor of work engagement in nurses [73].

Third, organizational support and resilience, resource factors from job and personal domain respectively, both negatively affected burnout. Similarly, a study of healthcare professionals in Saudi Arabia found a significant negative association between resilience and burnout [74]. Healthcare professionals with higher levels of resilience had lower levels of burnout. Zhou et al. examined burnout and wellbeing in healthcare workers during the post-pandemic period, and perceived organizational support was negatively associated with burnout [75].

Fourth, among the resource factors only organizational support (job resources) was positively related to work engagement and negatively to burnout. Organizational support has been recognized as a key factor in promoting work engagement [9, 76]. During the pandemic organizational support was important to ensure a safe workplace through the provision of personal protective equipment (e.g., gloves and masks) and developing policies to prevent the spread of the virus. In terms of personal resources (i.e., resilience), they were significantly and negatively related to burnout, but there was no significant influence on work engagement. Individuals with lower levels of resilience experience more physical and psychological effects that may lead to burnout, characterized by depersonalization, emotional exhaustion, and lack of personal accomplishment [35]. Resilience was also a relevant addition to this model since it positively influences the health impairment process. A systematic review and meta-analysis of interventions to build resilience in nurses found improvements in wellbeing [77].

Finally, burnout was negatively associated with wellbeing, while work engagement was positively associated with wellbeing. Burnout had a much stronger effect on wellbeing than work engagement. Workload impacted burnout more than any other factor. Furthermore, the resources were unable to compensate for the demands, leading to burnout and affecting wellbeing. This suggests the importance of focusing on workload during a pandemic to produce positive outcomes.

## Theoretical implications

Our results were generally aligned with the JD-R model and deepened our understanding of the relationship among demands, resources, and healthcare worker wellbeing in a crisis. However, there were two interesting findings. First, there was a non-significant relationship between workload and work engagement. This is consistent with the findings in a meta-analysis of longitudinal studies [78]. However, in the work of Kato et al. [43], job demands, specifically quantitative workload, were consistently negatively associated with work engagement in nurses in Japan. Similarly, a study among home health care nurses in Belgium found a negative association between workload and work engagement [47]. While some studies that extended JD-R with the job demands-work engagement relationship suggested a negative relationship due to job demands being perceived as burdens [47, 79], other studies found a positive relationship when job demands were perceived as challenging [79, 80].

Secondly, there was a non-significant relationship between resilience and work engagement. Demerouti and Bakker purported that high resilience leads employees to be more engaged in their work [34]. Ceschi et al. found resilience was positively associated with job demands and negatively associated with exhaustion, acting as a "psychological shield" [81]. During the pandemic, various researchers focused on building resilience to buffer the psychological stresses and support healthcare worker wellbeing [2, 52, 82–86].

## Practical implications

The findings of this study also have practical implications. For example, the significant relationship between workload and burnout highlights that healthcare organizations should consider strategies to manage and distribute workloads effectively, such as limiting shift hours,

hiring additional staff, and implementing efficient scheduling system to prevent burnout. At the same time organizational support was found as a crucial burnout reducing factor. Healthcare institutions should ensure availability of support as well as the awareness of such support. The support could include personal protective equipment and up-to-date information. Support system to address mental health needs can be beneficial too. Furthermore, while the study did not find a direct relationship with work engagement, it has been found associated with lowering levels of burnout. Healthcare organizations can arrange for training programs focusing on building resilience such as stress management workshops and peer support groups to help healthcare workers cope better with demand of the work. Lastly, loneliness as a personal demand was found to increase burnout and decrease work engagement. The healthcare institutions may upgrade their efforts to enhance social connections and provide opportunities to interact.

## Limitations

There are limitations to our findings. First, because a cross-sectional study design was used, it is impossible to draw firm conclusions on causal relationships. Secondly, while we added the personal demand of loneliness and the personal resource of resilience, other constructs may have been better predictors of wellbeing, as suggested by low item loadings of three resilience indicators. Many other potential constructs to be used in the model can be found in the literature [38, 49]. Third, a convenience sample was used, which can increase sampling bias. Next, participation was voluntary, which may lead to selection bias. Generalizability may not extend to other countries due to the unique setting in KSA with a large percentage of expatriate workers. Finally, healthcare workers in all departments from across the organization were included, and each professional discipline may have a JD-R model specifically suited for it.

## Conclusion

Previous research on healthcare workers has shown poor wellbeing to be associated with poor patient safety. Therefore, research on factors associated with wellbeing can be helpful in promoting healthcare wellbeing and in providing safe, high-quality patient care. Our study found workload was a major contributor to burnout, which had a large effect on wellbeing. Potential strategies should be implemented to reduce the workload by limiting shift times/overtime and creating a pool of additional staff. Although increasing social connections is difficult in a pandemic, every attempt should be made to promote camaraderie among team members. Given our finding that organizational support was negatively associated with burnout, it seems logical that every healthcare organization has some capacity and a good reason to identify new and improved ways of supporting employees. While we did not find a direct relationship between resilience and work engagement, we did find that higher levels of resilience were associated with lower levels of burnout. Therefore, it would be beneficial to introduce strategies that can help healthcare workers build resilience. Job demands and resources may have an important impact on healthcare wellbeing in a crisis situation. Our findings support the use of the JD-R model in this context.

## Author Contributions

**Conceptualization:** Consuela Cheriece Yousef, Ali Farooq, Gigi Amateau, Laila Carolina Abu Esba, Keisha Burnett, Omar Anwar Alyas.

**Data curation:** Consuela Cheriece Yousef, Ali Farooq, Gigi Amateau, Laila Carolina Abu Esba, Keisha Burnett, Omar Anwar Alyas.

**Formal analysis:** Ali Farooq.

**Methodology:** Consuela Cheriece Yousef, Ali Farooq, Gigi Amateau, Laila Carolina Abu Esba, Keisha Burnett, Omar Anwar Alyas.

**Software:** Ali Farooq.

**Validation:** Ali Farooq.

**Writing – original draft:** Consuela Cheriece Yousef.

**Writing – review & editing:** Consuela Cheriece Yousef, Ali Farooq, Gigi Amateau, Laila Carolina Abu Esba, Keisha Burnett, Omar Anwar Alyas.

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
