## [Decision Letter · Decision Letter 0]

17 Jan 2024

PONE-D-23-27017The Effect of Job and Personal Demands and Resources on Healthcare Workers' WellbeingPLOS ONE

Dear Dr. Yousef,

Thank you for submitting your manuscript to PLOS ONE. After careful consideration, we feel that it has merit but does not fully meet PLOS ONE’s publication criteria as it currently stands. Therefore, we invite you to submit a revised version of the manuscript that addresses the points raised during the review process.

We look forward to receiving your revised manuscript.

Kind regards,

Othman A. Alfuqaha, Ph.D.

Academic Editor

PLOS ONE

Journal Requirements:

Did you know that depositing data in a repository is associated with up to a 25% citation advantage (https://doi.org/10.1371/journal.pone.0230416)? If you’ve not already done so, consider depositing your raw data in a repository to ensure your work is read, appreciated and cited by the largest possible audience. You’ll also earn an Accessible Data icon on your published paper if you deposit your data in any participating repository (https://plos.org/open-science/open-data/#accessible-data).

3. We noticed you have some minor occurrence of overlapping text with the following previous publications (but not limited to these), which needs to be addressed:

1) Yousef, C.C., Salgado, T.M., Farooq, A., Burnett, K., McClelland, L.E., Abu Esba, L.C., Alhamdan, H.S., Khoshhal, S., Aldossary, I., Alyas, O.A. and DeShazo, J.P., 2022. Predicting health care providers' acceptance of a personal health record secure messaging feature. Applied Clinical Informatics, 13(01), pp.148-160.

2) Yousef, C.C., Salgado, T.M., Burnett, K., McClelland, L.E., Alhamdan, H.S., Khoshhal, S., Aldossary, I., Alyas, O.A. and DeShazo, J.P., 2023. Perceived barriers and enablers of a personal health record from the healthcare provider perspective. Health Informatics Journal, 29(1), p.14604582231152190.

In your revision ensure you cite all your sources (including your own works), and quote or rephrase any duplicated text outside the methods section. Further consideration is dependent on these concerns being addressed.

4. We note that your Data Availability Statement is currently as follows: "All relevant data are within the paper and the supporting information files."

5. Please ensure that you include a title page within your main document. You should list all authors and all affiliations as per our author instructions and clearly indicate the corresponding author.

6. Please amend your manuscript to include your abstract after the title page.

**Additional Editor Comments:**

Dear Authors,

Thank you for your insightful article. Overall, the aim and methodology are well-structured, and the results are effectively presented. However, I have identified some areas that could benefit from improvement:

Abstract:

Clarify the discrepancy between mentioning a single hospital and describing a multicenter study.

Provide more details on the sample size recruitment procedure, such as the use of convenience sampling.

Introduction:

Explicitly state the unknown aspects covered by your study in each paragraph.

Clearly articulate the specific individual and job characteristics you aim to investigate.

Include the main aim of the study, as outlined in the JD-R model, in the abstract section.

Methods:

Elaborate on how you ensured that healthcare professionals from other areas of KSA responded to your recruitment efforts.

Specify the sampling method used.

Clarify whether the study includes multiple cities, with a mention of Dammam.

Indicate the language used in the survey (Arabic or English).

Results:

Provide explanations for symbols in Table 1.

Mention abbreviations before their first use.

Include significance values (P<0.001**) in Table 3.

Discussion:

Begin the discussion with a paragraph highlighting the key results without repeating the aim.

Expand on the unique aspects of KSA healthcare professionals contributing to burnout or other studied factors.

Ensure clarity and readability of figures.

Please consider these suggestions for enhancing the article. Your attention to these details will further strengthen the overall quality of your work.

Best regards,

Othman A. Alfuqaha

Reviewers' comments:

Reviewer's Responses to Questions

**Comments to the Author**

1. Is the manuscript technically sound, and do the data support the conclusions?

Reviewer #1: Yes

Reviewer #2: Yes

2. Has the statistical analysis been performed appropriately and rigorously? 

Reviewer #1: Yes

Reviewer #2: Yes

3. Have the authors made all data underlying the findings in their manuscript fully available?

Reviewer #1: Yes

Reviewer #2: Yes

4. Is the manuscript presented in an intelligible fashion and written in standard English?

Reviewer #1: Yes

Reviewer #2: Yes

5. Review Comments to the Author

Reviewer #1: Overall it is good research and contribution to the existing body of literature based on empirical analysis and comprehensive discussion, however the implications of the study are not discussed in explicit way.

Reviewer #2: Respected,

The current manuscript aims to analyze the relationship of various individual and job

characteristics with regards to employee wellbeing in a population of healthcare workers in a

government hospital in the Kingdom of Saudi Arabia.

The paper should be accepted for publication after some clarifications:

- please define the categories of health workers (healthcare professionals, health associate professionals, personal care

workers, health management and support personnel, and health service providers not elsewhere

classified) and their workplace tasks and activities,

- it is good to list the hypotheses in a similar fashion as they are shown in the other parts of the manuscript (workload, loneliness, organization support, and resilience as antecedents; burnout and work engagement as mediators; and wellbeing as a dependent variable),

- please describe the importance of the findings for the practical work and implementation of preventive activities.

6. PLOS authors have the option to publish the peer review history of their article (what does this mean?). If published, this will include your full peer review and any attached files.

Reviewer #1: No

Reviewer #2: **Yes: **Dragan Mijakoski

---

## [Author Response · Author response to Decision Letter 0]

26 Mar 2024

Additional Editor Comments

7. Abstract: Clarify the discrepancy between mentioning a single hospital and describing a multicenter study. Provide more details on the sample size recruitment procedure, such as the use of convenience sampling

Response: The multicenter was removed and convenience sampling was added. 

8. Explicitly state the unknown aspects covered by your study in each paragraph. Clearly articulate the specific individual and job characteristics you aim to investigate. Include the main aim of the study, as outlined in the JD-R model, in the abstract section.

Response: Abstract has been rewritten to address the issue raided in the comment.

9. Methods: Elaborate on how you ensured that healthcare professionals from other areas of KSA responded to your recruitment efforts. Specify the sampling method used. Clarify whether the study includes multiple cities, with a mention of Dammam. Indicate the language used in the survey (Arabic or English).

Response: Additional details were added in the “Recruitment” section: “Data were collected through an anonymous online survey in English using Google Forms between February and March 2022. The survey link was shared through email invitations sent through the organization’s listserv for healthcare workers in all cities (Dammam, Al Ahsa, Riyadh, Madinah, and Jeddah) and through WhatsApp messages sent to personal contacts of two of the authors.”

Under “Study Participants” the following statement was added: “Convenience sampling was used to recruit participants from all cities.”

10. Results: Provide explanations for symbols in Table 1. Mention abbreviations before their first use. Include significance values (P <0.001**) in Table 3.

Response: Table 1 contains only two symbols ‘N’ and ‘n’. ‘N’ denotes sample size, whereas, ‘n’ donates sub-samples in each group. We have added a note to Table 1 to explain these.

The abbreviations used in Table 3 are described in the note after the table. We have added p value to the title of Table 3. 

11. Discussion: Begin the discussion with a paragraph highlighting the key results without repeating the aim. Expand on the unique aspects of KSA healthcare professionals contributing to burnout or other studied factors. Ensure clarity and readability of figures

Response: In line with the comment, we have restructured the discussion into sub-sections of key findings, theoretical and practical implications, and limitations.

Reviewer 1

12. Overall it is good research and contribution to the existing body of literature based on empirical analysis and comprehensive discussion, however the implications of the study are not discussed in explicit way.

Response: We have explained theoretical and practical implications.

Reviewer 2

13. The current manuscript aims to analyze the relationship of various individual and job characteristics with regards to employee wellbeing in a population of healthcare workers in a government hospital in the Kingdom of Saudi Arabia. The paper should be accepted for publication after some clarifications: - please define the categories of health workers (healthcare professionals, health associate professionals, personal care workers, health management and support personnel, and health service providers not elsewhere classified) and their workplace tasks and activities,

Response: The categories were clarified, and a reference was added to the WHO classification of healthcare workers.

14. It is good to list the hypotheses in a similar fashion as they are shown in the other parts of the manuscript (workload, loneliness, organization support, and resilience as antecedents; burnout and work engagement as mediators; and wellbeing as a dependent variable)

Response: We have updated the hypotheses to look coherent in the hypotheses section

15. Please describe the importance of the findings for the practical work and implementation of preventive activities

Response: We have now included both the theoretical and practical implications of the study.

---

## [Editor Report · Decision Letter 1]

1 May 2024

The effect of job and personal demands and resources on healthcare workers' wellbeing: A cross-sectional study

PONE-D-23-27017R1

Dear Dr.

We’re pleased to inform you that your manuscript has been judged scientifically suitable for publication and will be formally accepted for publication once it meets all outstanding technical requirements.

Kind regards,

Othman A. Alfuqaha, Ph.D.

Academic Editor

PLOS ONE

Additional Editor Comments (optional):

Congratulations.
---

## [Editor Report · Acceptance letter]

9 May 2024

PONE-D-23-27017R1 

PLOS ONE

Dear Dr. Yousef, 

I'm pleased to inform you that your manuscript has been deemed suitable for publication in PLOS ONE. Congratulations! Your manuscript is now being handed over to our production team.

Kind regards, 

on behalf of

Dr. Othman A. Alfuqaha 

Academic Editor

PLOS ONE